# The Progress of the New South Wales Aboriginal Oral Health Plan 2014–2020: A Scoping Review

**DOI:** 10.3390/healthcare10040650

**Published:** 2022-03-30

**Authors:** Ashwaq Maqbool, Charlotte Marie Selvaraj, Yinan Lu, John Skinner, Yvonne Dimitropoulos

**Affiliations:** 1Faculty of Medicine and Health, School of Public Health, The University of Sydney, Sydney, NSW 2006, Australia; cdac3196@uni.sydney.edu.au (C.M.S.); yilu0253@uni.sydney.edu.au (Y.L.); 2Poche Centre for Indigenous Health, Faculty of Medicine and Health, The University of Sydney, Sydney, NSW 2006, Australia; john.skinner@sydney.edu.au (J.S.); yvonne.dimitropoulos@sydney.edu.au (Y.D.)

**Keywords:** Aboriginal and Torres Strait Islander people, oral health, NSW, scoping review, oral health plan

## Abstract

There are major disparities in oral health between Aboriginal and Torres Strait Islander and non-Aboriginal and Torres Strait Islander people. The New South Wales (NSW) Aboriginal Oral Health Plan 2014–2020 was developed to improve the oral health of Aboriginal people. This scoping review describes programs that have been undertaken to implement the NSW Aboriginal Oral Health Plan 2014–2020. The methodology by Arksey and O’Malley was used to guide this review. Academic and grey literature were searched using a structured Medline, Lowitja and advanced Google searches. Articles were included if they aligned with the strategic directions of the Plan. Key information, including the aims of the study, methodology and results were recorded in a template on Microsoft Excel software. A total of 31 articles were included in this review. This included 25 articles from the academic literature and six initiatives from the grey literature. Included articles were categorised according to the six strategic directions in the NSW Aboriginal Oral Health Plan. Four studies were related to the first strategic direction, six related to strategic direction two, four related to strategic direction three, six initiatives related to strategic direction four, five related to strategic direction five, and eight related to strategic direction six. While there has been significant progress in achieving the strategic directions of the NSW Aboriginal Oral Health Plan, there is scope for continued collaboration between oral health service providers, universities and Aboriginal communities to improve oral health outcomes for Aboriginal people in NSW.

## 1. Introduction

There are major disparities in oral health between Aboriginal and Torres Strait Islander and non-Aboriginal and Torres Strait Islander people. Many Aboriginal and Torres Strait Islander people have limited access to culturally competent oral health services and oral health promotion [1]. Aboriginal and Torres Strait Islander children and teenagers are overrepresented in requiring removal or restorations of teeth under general anaesthesia. As a result, the rate of potentially preventable hospitalization is high (13.3 per 1000 people for Aboriginal and Torres Strait Islander people compared to 9.3 per 1000 for non-Indigenous people). Ha et al. (2016) found that Aboriginal and Torres Strait Islander children who are 6–9 years develop, on average, 1.3 decayed, missing or filled tooth surfaces (DMFS) compared to 0.7 non-Aboriginal and Torres Strait Islander children [1].

To address these disparities, the New South Wales (NSW) Ministry of Health developed the NSW Aboriginal Oral Health Plan 2014–2020 [2]. It describes a range of key strategic directions to improve oral health outcomes for Aboriginal people. It is the first strategic plan published and funded by NSW Health focused on improving the oral health of Aboriginal people. The plan aims to build collaborative relationships with key partners such as Aboriginal communities, the Australian Dental Association, Local Health Districts and Universities to support the provision of culturally appropriate and safe oral health services and oral health promotion for Aboriginal people [2].

Often, the progress of state and national health plans is not measured or reviewed, and rarely via peer-reviewed literature. Hence, this study aims to conduct a scoping review to describe programs that have been implemented as part of the NSW Aboriginal Oral Health Plan and to analyse progress made against the plan and its impact in NSW over the last five years. Conducting this scoping review also aims to identify gaps to be addressed by future programs as well as programs that show promise in improving oral health outcomes for Aboriginal people for future scale-up.

## 2. Materials and Methods

The scoping review methodology by Arksey and O’Malley was used to guide this review [3]. Both academic and grey literature was searched to conduct this review. The academic literature was searched using structured Medline (see Appendix A) and Lowitja searches, and an advanced Google search was used for grey literature. A university Assistant Librarian (JH) was consulted to develop the search strategy for the Medline search engine. A review protocol was not registered for this study.

### 2.1. Eligibility Criteria

The six strategic directions of the NSW Aboriginal Oral Health Plan were used to guide the inclusion and exclusion criteria for this review.

The strategic directions of the NSW Aboriginal Oral Health Plan include:Increase access to fluoridated water and fluoride programs to assist in the reduction of dental caries.Develop and implement sustainable oral health promotion and prevention programs.Improve access to appropriate dental services for Aboriginal people in NSW in culturally safe environments.Develop and implement sustainable programs to increase the number of Aboriginal people in the oral health workforce in NSW.Strengthen the capacity of the existing and future health and oral health workforce to provide appropriate oral health care in Aboriginal communities.Improve oral health for Aboriginal people through supported action-oriented research and improved oral data collections for evaluating both service delivery and oral health outcomes.

Articles yielded from each search were screened independently by C.M.S., A.M. and Y.L. (Medline (C.M.S.); Lowitja (A.M.); Advanced Google (Y.L.)) for inclusion in this study. Articles were included if they aligned with any of the six strategic directions of the Plan [2].

Articles needed to be published in English during the life of the Plan (between 2014 and 2020). Once screened, all authors discussed and agreed upon the final inclusion of articles.

Authors C.M.S., A.M. and Y.L. independently charted key information on included articles using a template on Microsoft Excel Software. This included publication year, location, aim, methodology and results. The authors did not seek additional information from authors of included publications. The template used was used in prior published scoping reviews by authors J.S. and Y.D.

### 2.2. Data Synthesis

The next stage of the review involved assigning included articles to the strategic direction of the plan. A.M., C.D. and Y.L. independently assigned each article to the six strategic directions of the plan. All authors reviewed the allocation of articles to a strategic direction.

### 2.3. Ethical Consideration

There was no patient involvement in this study as information was publicly available. Therefore, ethical approval was not required.

## 3. Results

A total of 31 articles were included in this review. This included 25 articles from the academic literature and six initiatives from the grey literature. Of the 25 articles retrieved from the academic literature search, nine were from the Medline search and sixteen from the Lowitja search. Figure 1 shows the number of articles yielded from the initial search, screened, assessed for eligibility and included in the review, with reasons for exclusions at each stage.

Table 1 shows the studies included in this article from the Medline and Lowitja searches.

### 3.1. Increase Access to Fluoridated Water and Fluoride Programs to Assist in the Reduction of Dental Caries

Four studies were found related to increasing access to fluoride varnish programs for Aboriginal people, including children. The regular application of fluoride varnish is an effective and safe strategy known to improve oral health by reducing the risk of dental decay in vulnerable communities [7]. A co-designed school-based fluoride varnish program run over a year with local Aboriginal communities in 2017, by Dimitropoulos et al. (2020), utilised Oral Health Therapists to apply fluoride varnish four times per year for Aboriginal school children in remote and rural NSW. The program provided over 65% of students with at least three applications of fluoride [7]. The use of Aboriginal dental assistants was recommended by the study, to help ensure that oral health promotion programs are tailored to the community needs. This first pilot program demonstrated this model can encourage the development of state-wide fluoridation varnish programs to reduce the risk of dental decay in Aboriginal children, especially in communities where children are at higher risk of developing dental caries. Continuing this research, Dimitropoulos et al. (2019) developed a protocol enabling Aboriginal dental assistants to apply fluoride varnish every three months to Aboriginal school children [14]. Enabling the Aboriginal paraprofessional oral health workforce to apply fluoride varnish can increase access to fluoride programs and improve the oral health of Aboriginal children in NSW [14]. Skinner et al. (2020) costed the scale-up child fluoride varnish programs in NSW based on this model [15]. The study found this standardised approach for school-based programs to be feasible, with the costs of four applications of fluoride varnish a year mainly covered by Child Dental Benefits Schedule [15]. Publicly funded dental services and oral health can play in role in increasing access to preventive dental care for Aboriginal communities. A National Fluoride Varnish Workshop held in 2018 considered key steps that needed to be taken to expand current fluoride varnish programs in Australia. The workshop emphasised the need for further standardised training for the paraprofessional oral health and non-dental workforce to apply fluoride varnish [19]. Further issues raised at the workshop included how the government could expand the current non-dental workforce, and their ability to access funding via the Child Dental Benefit Schedule for fluoride varnish [19]. In supporting fluoride varnish programs as key public health measures in the NSW Aboriginal Oral Health Plan, this study acknowledges that the continuing growth of these programs is reliant on a scale-up of the current workforce to support these expanding initiatives, utilising both a dental and non-dental workforce [19].

### 3.2. Develop and Implement Sustainable Oral Health Promotion and Prevention Programs

Six studies were found related to the development and implementation of sustainable oral health promotion programs. One study by Dimitropoulos et al. (2014) collaborated with Aboriginal communities in Central Northern NSW to design and implement a community-owned oral health promotion program [17]. The collaboration included determining the oral health needs of Aboriginal children, oral health knowledge of parents and guardians and enablers and barriers towards developing an oral health promotion program as perceived by school staff. The steps were achieved by a dental examination and oral health questionnaires. As a result, 35% of children did not own a toothbrush, 24% did not use fluoride toothpaste, and parents and guardians had limited oral health knowledge [17]. This article showed that developing a sustainable oral health prevention and promotion program is crucial and urgent to reduce the burden of dental caries and improve oral health awareness. Based on the collaboration, a protocol for a community-led oral health promotion program was co-designed with the Aboriginal communities and consisted of four components: daily in-school toothbrushing, distributing free fluoride toothpaste, community oral health education and installing cold water fountains to implement a school’s water bottle supply program to increase water consumption [18]. It was implemented between 2016–2018 in three schools in Central Northern NSW. As a result, there was a significant reduction in dental caries and plaque scores and a substantial change in oral health behaviour. The overall decayed, missing, filled teeth (dmft/DMFT) score was reduced from 5.31 in 2014 to 4.13 in 2018, and there was a significant reduction in untreated tooth decay: 4.27 in 2014 compared to 2.27 in 2018 [10]. There was an increase in children who brushed their teeth daily (36% in 2018 compared to 13% in 2014), largely due to the distribution of free toothbrushes and fluoride toothpaste [10]. In terms of water consumption, 84% of children filled their water bottles daily from the new refrigerated water fountains, and 33% of children consumed sweetened beverages regularly in 2018 compared to 64% in 2014 [10]. The school toothbrushing component was also evaluated by exploring the perspectives of teachers who supervised children’s toothbrushing [8]. The thematic analysis demonstrated that the success and sustainability of the program was attributed to school ownership and training local Aboriginal people and existing school staff to deliver the program. This program demonstrated co-designed and community-led oral health promotion can improve oral health outcomes for Aboriginal children [10]. Furthermore, this program demonstrated sustainability and increased oral health awareness in the Aboriginal community [8]. This was largely due to empowering the local Aboriginal community to lead the design and implementation of the oral health promotion program to improve oral health outcomes for their community [8].

A graduate oral health therapist program to support dental service delivery and oral health promotion in Aboriginal communities was established in 2016, known as the “Dalang project” [28]. Graduate oral health therapists were employed in Aboriginal Community Controlled Health Services (ACCHS) to deliver oral health services and oral health promotion. The program aimed to improve the oral health of Aboriginal children, increase employment for Aboriginal people as dental assistants, provide a positive experience for new graduates in Indigenous health and strengthen the evidence of health promotion and prevention in Aboriginal communities. The study by Skinner et al. (2021) evaluated the Dalang Project [28]. The majority of oral health therapists who participated in the program recommended the program to future graduates. Oral health therapists reported learning to engage with Aboriginal communities and build cultural competence skills to provide culturally competent oral health services and develop oral health promotion programs. The types of oral health promotion delivered as part of the Dalang Project included in-school toothbrushing programs, distribution of fluoride toothpastes, refrigerated water fountain installation, dental health education and regular application of fluoride. The program helped engage health therapists with the Aboriginal community and helped deliver culturally safe oral health promotion and oral health services [28].

The “Smile not Tears” initiative was another oral health promotion program for Aboriginal children that was developed to address dental disease in Aboriginal communities [16]. The program aimed to reduce the prevalence of early childhood caries among Aboriginal children. It was an evidence-based program that trained and supported Aboriginal Health Workers (AHWs), who in turn trained local parents and guardians by providing simple information about early childhood caries; this was supported by oral examinations to detect carious lesions [16]. It was ensured that key members of Aboriginal communities were involved in the design of the program and that all aspects of the research were culturally acceptable. The assessment aimed to evaluate the effectiveness of the Smile Not Tears program [16]. As a result, 97% of children in the test group were caries-free compared to 65% in the control group. Therefore, the program showed a positive impact on reducing caries prevalence in Aboriginal children, which shows the successful implementation of the sustainable oral health promotion and prevention program [16].

### 3.3. Improve Access to Appropriate Dental Services for Aboriginal People in NSW in Culturally Safe Environments

Three articles were found relating to improving access to appropriate dental services for Aboriginal people. A study by Gwynne et al. (2016) described a sustainable model of oral health care that was developed to deliver reliable, high-quality oral health services to Aboriginal children and adults using a Collective Impact model [27]. The program developed a shared agenda between local ACCHS, the Poche Centre for Indigenous Health at the University of Sydney and the local Aboriginal community to deliver dental services in schools and local community health centres using mobile and portable equipment [27]. The model proved to be effective in providing culturally appropriate dental services. Further, a mobile denture service that provides dentures in a short time frame was also established by the Poche Centre for Indigenous Health [21]. The mobile denture clinic is operated by an Aboriginal dental technician and prosthodontists and a team of local dentists, oral health therapists and local Aboriginal dental assistants. As a result, the novel denture service showed a significant improvement in the recipients’ quality of life, mental health and oral health [21].

Another program to increase access to dental services for Aboriginal people was provided by the Royal Flying Doctor Service [4]. It provided preventive, diagnostic, restorative, endodontic and general dental services to various communities in NSW using mobile dental clinics. The purpose of the program was to expand dental services to areas with inadequate service provision and encourage stakeholders to develop policies that enhance the adequacy and availability of dental services. As a result, the program provided dental services to patients in remote areas of NSW and improved accessibility to dental services for people in these regions [4].

### 3.4. Develop and Implement Sustainable Programs to Increase the Number of Aboriginal People in the Oral Health Workforce in NSW

Six initiatives were found in the grey literature related to the development and implementation of sustainable workforce programs. One of the initiatives found was a signed Memorandum of Understanding (MOU) between the Indigenous Allied Health Australia (IAHA) and Indigenous Dentists Association of Australia (IDAA) [29]. The second initiative was the expansion of the National Aboriginal and Torres Strait Islander Health Academy Model [30]. The remaining four initiatives were scholarships to encourage Aboriginal people to pursue tertiary studies in oral health [31,32,33,34].

The MoU between the IAHA and IDAA was signed on 21 May 2018. It was developed to support Aboriginal dentists and dental students and improve the health and well-being of Aboriginal and Torres Strait Islander people [29]. The MoU was a welcomed partnership, as IDAA is committed to supporting Aboriginal dentists and dental students and improving oral health care for Aboriginal people, and members of IAHA include Aboriginal dentists [29]. This initiative of the IAHA and IDAA aims to provide ongoing support to ensure culturally competent oral health care can be delivered.

The National Aboriginal and Torres Strait Islander Health Academy Model is a community-led learning model developed by the IAHA to support Aboriginal students to complete year 12 with a Certificate III in Allied Health [30]. The first academy was established in 2018. Building on the success of the model, the IAHA expanded the academy to other states including NSW. IAHA members, including Aboriginal dentists, have played an important role in providing guidance to academy students, resulting in an interest in oral health among students [30]. This model may encourage Aboriginal students completing year 12 to pursue oral health-related occupations and can increase the number of Aboriginal people in the oral health workforce in NSW.

The four scholarships identified support and encourage Indigenous students to pursue tertiary studies in oral health and help students successfully complete their studies [31,32,33,34]. The four scholarships are summarised in Table 2.

### 3.5. Strengthen the Capacity of the Existing and Future Health and Oral Health Workforce to Provide Appropriate Oral Health Care in Aboriginal Communities

Four articles were found related to strengthening the capacity of the oral health workforce to provide appropriate oral health care in Aboriginal communities. The six initiatives related to increasing the number of Aboriginal people in the oral health workforce were also applicable to this strategic direction. The MoU signed between the IAHA and IDAA enables cooperation between the associations to improve the oral health of Aboriginal people and support Aboriginal dentists and dental students [28]. This support enables Aboriginal dentists and dental students to improve their abilities to provide appropriate oral health care in Aboriginal communities. The National Aboriginal and Torres Strait Islander Health Academy model can provide Aboriginal year 12 students with an interest in oral health support to pursue a career in oral health. This may be conducive to strengthening the capabilities of the future Aboriginal oral health workforce. The four scholarships noted in Table 1 are all awarded to Indigenous students completing tertiary education in oral health to help them complete their studies [31,32,33,34], strengthening the capacity of the future oral health workforce.

A qualitative study was found that documented the experience of dental clinicians who relocated to rural and remote NSW to provide dental services in Aboriginal communities [24]. The study found that providing personal and professional support to the relocated clinicians led to their improved professional skills and personal development. Living and working in the community also improved their cultural capabilities [24]. Community engagement and professional support can strengthen the capacity of oral health workers to provide appropriate oral health care for Aboriginal people.

Two studies were found examining the integration of Aboriginal cultural competence training with dental courses at the University of Sydney School of Dentistry [5,6]. The first case study conducted by Forsyth et al. (2020) identified intervention strategies to improve the cultural competence of students and conducted a baseline analysis of cultural competence curriculum practice [5]. It also identified obstacles and driving factors to improve the Aboriginal cultural competence of dental students. The study found that in dental education, Aboriginal cultural competence requires strict management, sufficient teacher resources and effective education strategies to increase students’ knowledge, understanding and skills to improve the cultural competence abilities of the future oral health workforce [5]. The second study by Forsyth et al. (2019) conducted semi-structured interviews with academics from the Doctor of Dental Medicine (DMD) and Bachelor of Oral Health (BOH) programs at the University of Sydney [6]. The study aimed to define and explore the current Aboriginal cultural competence curriculum, determine the driving factors and obstacles to integrate the cultural competence curriculum and determine strategies to help students improve their cultural competence [6]. It found that improving the cultural competence of students requires educational and philosophical changes [6]. It needs to include the informed history of Australian Aboriginal people, to allow students to immerse themselves in the Aboriginal and Torres Strait Islander culture, as well as reflection on these experiences [6]. Improving the cultural competence of people completing tertiary education in oral health and medicine can promote the provision of culturally appropriate oral and general health care in Aboriginal communities.

A study by Kong et al. (2020) explored the views and experiences of Australian AHWs on the oral health of Aboriginal women during pregnancy and aimed to develop a new, culturally safe oral health care model for Aboriginal women during pregnancy [9]. The study found that it is necessary to train AHWs and Family Partnership Workers in oral health to develop an effective oral health care model for Aboriginal women during pregnancy [9]. Oral health training for AHWs will enable provision of appropriate oral health care for Aboriginal women during pregnancy and in turn may also prevent dental caries in Aboriginal children.

### 3.6. Improve Oral Health for Aboriginal People through Supported Action-Oriented Research and Improved Oral Data Collections for Evaluating Both Service Delivery and Oral Health Outcomes

Eight articles were found related to action-oriented research and improved oral data collection for evaluation service delivery and oral health outcomes. Kong et al. (2021) explored if oral health was an important consideration for Aboriginal women in NSW during pregnancy [22]. This study employed a qualitative descriptive methodology with interviews analysed using thematic analysis [22]. The study found that effective oral health promotion should involve key healthcare providers, including AHWs, to facilitate culturally safe models of oral health care and oral health promotion [22]. This study fostered the engagement of the community to address disparities in oral health through research, finding that that oral health outcomes can be improved through identifying new models of care that meet the unique needs of vulnerable populations [22].

A longitudinal cohort study by George et al. (2018) looking at the oral health behaviours and fluid intake of Aboriginal children in South-Western Sydney found limited uptake of dental services among urban Aboriginal children [20]. The study highlighted the need for targeted oral health promotion to improve engagement with oral health services [20]. This research provides insight on social factors and the experiences of oral health for Aboriginal youth to allow culturally appropriate oral health services promotion to be delivered.

A mixed methods study by Campbell et al. (2015) looking at online surveys and semi- structured interviews with ACCHSs investigated the oral health care experiences that inform policy and program decision making [11]. The study found that ACCHSs would be greater strengthened by acknowledging their expertise and establishing long-term funding models that are responsive to the needs of Aboriginal communities [11]. Improved data collection, such as in this study on oral health care experiences, will continue to support the goal of expanding oral health care services, by promoting the translation of research findings into practice and use.

A study by Gwynne et al. (2017) compared two public oral health models of care for Aboriginal people in NSW. Model A was a “Fly in Fly Out” and Model B was a local community-led model [13]. Two years’ worth of Dental Weighted Activity Units and funding was compared [13]. The study found that Model B (community-led model) provided significantly more services for less financial resources (47% more treatment at 25.2% of the cost) [13]. Another study by Gwynne et al. (2021), compared the scope of practice between the two models, using the Australian Dental Association schedule of services [26]. The study found that Model A (Fly in Fly Out model) provided more complex dental care, whilst Model B (community-led model) provided more preventive dental care [26]. Both these studies support oral health research to recognise opportunities to increase the effectiveness of current oral health care service delivery through identifying improved models of care.

A cross-sectional study by Smith et al. (2015) examined the oral health status of 196 Aboriginal children living in NSW [25]. The study found a high prevalence of untreated dental caries (88.3%) in Aboriginal children living in remote NSW [25]. Oral health epidemiology is particularly important in the NSW Aboriginal Oral Health Plan in describing the oral health status of Aboriginal people, as this helps to quantify goals to achieve improvements in oral health care delivery and services. Irving et al. (2017) examined the views of parents and children who accessed a community-led oral health service in Central Northern NSW in 2013–2014 [12]. The study found that the implementation of this community-led oral health service was well utilised and that these collaborative approaches should be continued to implement targeted oral health promotion programs to encourage positive oral hygiene practices in Aboriginal communities [12].

A study conducted by Orr et al. (2021) sought to examine the use of a Child Dental Benefits Schedule among Aboriginal and non-Aboriginal children [23]. The study found lower levels of service utilisation for Aboriginal children, particularly for preventative services [23]. Limited access to preventive oral health services among Aboriginal children can lead to future inequalities in oral health and highlights the need for preventive oral health promotion programs and programs that facilitate access to preventive dental care, such as the regular application of fluoride varnish for Aboriginal children.

## 4. Discussion

This scoping review aimed to evaluate and assess progress against NSW Aboriginal Oral Health Plan 2014–2020 by searching the academic and grey literature. The articles found included studies collaborating with local Aboriginal communities to provide oral health promotion, providing culturally competent oral health services, action-orientated research to inform funding of service delivery models and plan oral health promotion programs, research to increase the cultural capabilities of the existing and future oral health workforce to provide appropriate dental care and initiatives providing Aboriginal people scholarships to support Aboriginal people pursuing tertiary studies in oral health.

There has been considerable development and implementation of sustainable oral health promotion programs for Aboriginal people, especially children living in rural and remote communities. Overall, oral health promotion programs co-designed with local Aboriginal communities have shown significant improvement in oral health outcomes for Aboriginal people, with multiple studies published assessing the effectiveness of co-designed oral health programs [10,16]. Many programs that were found trained local Aboriginal people to deliver the oral health promotion programs [8,10,14,16], and this approach proved sustainable and built the capacity of the local Aboriginal community. Implementing evidence-based and simple long-term strategies, such as the installation of refrigerated water fountains, distribution of toothbrushes and fluoride toothpaste and in-school toothbrushing, showed improved positive oral hygiene behaviours and increased consumption of water among Aboriginal children [10]. This review found that engaging local stakeholders, including schools, local ACCHS and professional institutions, increased program sustainability. These programs align with evidence-based oral health promotion guidelines and now need scaling-up nationally [35,36].

State-wide fluoride programs using the para-professional dental workforce such as Aboriginal dental assistants and the non-dental workforce have achieved considerable progress in aligning with the NSW Aboriginal Oral Health Plan as well as other oral health plans and position papers, including Australia’s National Oral Health Plan 2015–2024 [37,38,39]. Several studies in this review demonstrated the limited access to preventive dental care among Aboriginal children, highlighting the need for programs to improve access to preventive dental care, including fluoride varnish. Programs that provide regular applications of fluoride varnish in the school setting have shown to be feasible. Enabling the Aboriginal paraprofessional dental workforce and non-dental workforce, such as dental assistants and AHWs, to apply fluoride varnish can increase the accessibility and cultural safety of regular application of fluoride varnish. The success of these pilots has led to policy change nationally and inclusion of the “Apply Fluoride Varnish Skillset” in the Dental Assisting course [40,41,42].

Cultural safety is determined by Aboriginal and Torres Strait Islander people and is an important factor in accessing dental and general health care and a major focus of the Aboriginal Oral Health Plan [2]. The Australian Medical Association (AMA) defines cultural safety as ongoing critical reflection of the knowledge, skills, attitudes, practising behaviours and power differentials of health practitioners to delivering safe, accessible and responsive healthcare free of racism [43,44]. There has been a shift in the approaches of universities to begin embedding cultural competence training into oral health and medicine degrees to improve the cultural capabilities of the future oral and general health workforce to provide culturally safe care. Furthermore, several studies were identified that focused on the provision of culturally safe oral health services by developing partnerships with local Aboriginal communities to deliver oral health services and oral health promotion programs and training local Aboriginal people to deliver or assist in the delivery of these oral health services [13,26,27]. Further efforts and funding are now required to ensure these programs are appropriately scaled up [37].

There has also been considerable progress in increasing the Aboriginal dental workforce. These models as well as structured fluoride varnish programs have been supported by the AMA to address the oral health needs of Aboriginal and Torres Strait Islander people and increase the Aboriginal dental workforce. The AMA endorses their application by government and political leaders to deliver culturally safe, affordable and accessible oral health programs in Aboriginal communities [43] and improve access to preventive dental care for Aboriginal people, including children [44]. Collaboration between universities, ACCHSs and professional associations have made significant efforts to increase the number of Aboriginal dentists and oral health therapists by providing scholarships to Aboriginal people pursuing tertiary education in dentistry and oral health to assist completion of their studies. These efforts will undoubtedly increase the Aboriginal dental workforce. Supporting Aboriginal people to qualify as dentists, oral health therapists and dental assistants will in turn provide culturally safe dental care for Aboriginal people.

### Limitations

This literature review searched the academic and grey literature, capturing oral health promotion and prevention programs and initiatives aimed at improving the oral health of Aboriginal people reported in scientific journals and online. However, it is likely that several programs and initiatives that may have been implemented by ACCHSs, Local Health Districts or Aboriginal communities were not identified if they were not published in scientific journals or published online. Therefore, further research assessing progress against the NSW Aboriginal Oral Health Plan should involve input from ACCHSs and Local Health Districts on oral health promotion and prevention activities.

## 5. Conclusions

This review demonstrated that there has been significant progress in the strategic directions set out in the NSW Aboriginal Oral Health Plan 2014–2020. Co-designed programs highlight the need for oral health promotion and prevention programs and show their effectiveness and sustainability. Fluoride varnish application programs show the need for greater use of Aboriginal dental assistants and similarly qualified workforce to scale-up these programs. Collaboration between professional associations, universities and Aboriginal communities have led to effective programs and initiatives, increasing the oral health workforce and training dental students to deliver culturally safe dental services. While there has been significant progress in achieving strategic directions of the oral health plan, there is still scope for public oral health services and universities to collaborate with Aboriginal communities to continue to implement the strategic directions of the plan to improve the oral health of Aboriginal people in NSW.

## Figures and Tables

**Figure 1 healthcare-10-00650-f001:**
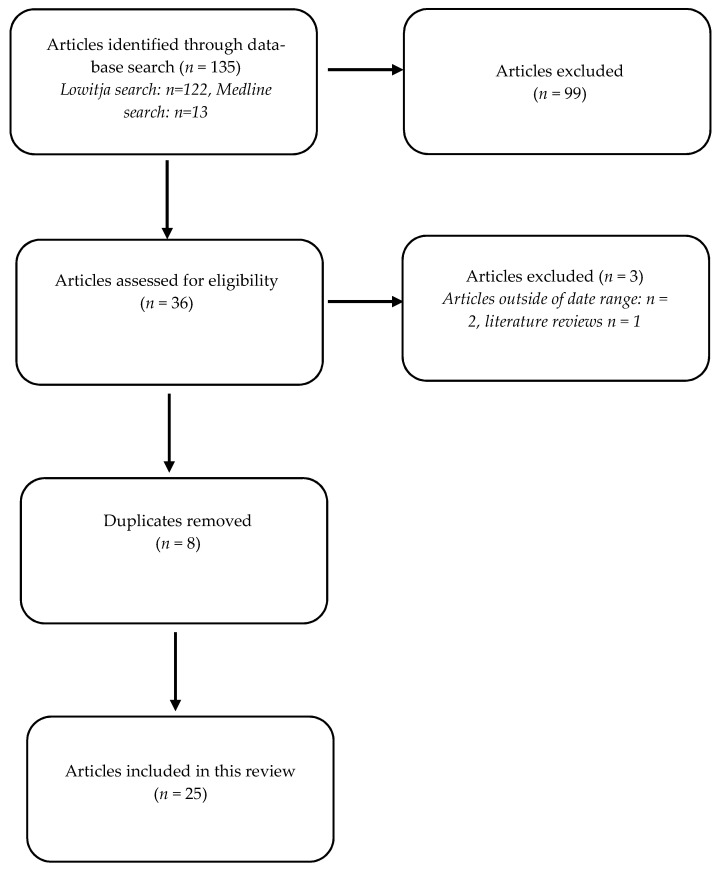
Study selection flowchart.

**Table 1 healthcare-10-00650-t001:** Summary table of included articles from Medline and Lowitja searches.

Author, Year	Aim	Methodology	Results	Strategic Direction
Gardiner et al., 2020 [4]	Determine service provision by Royal Flying Doctor Service	Dental services provided by the Royal Flying Doctors Services within rural and remote Australia between April 2017 to September 2018 were analysed	8992 service episodes, 3407 individual patients, 27,897 services completed	3
Forsyth et al., 2020 [5]	Examine the integration of Indigenous cultural competence in dental curricula	Four sources of data were analysed: systematic review online surveys two in-depth interviews with academics and students	Indigenous cultural model developed for dentistry education, Indigenous cultural competence in dentistry education requires governance, adequate resources and education	5
Forsyth et al., 2019 [6]	Define and explore current Indigenous cultural competence curricula in Dentistry and Medicine programmes at the University of Sydney	Semi-structured interviews were conducted with academics and students. Thematic analysis was conducted to analyse interviews	Six key themes emerged, including transfer of knowledge, barriers, importance, resources, proposed content and strategies.	5
Dimitropoulos et al., 2020 [7]	Determine if school fluoride varnish programs are feasible	A school fluoride varnish program was developed where fluoride varnish was applied at 3-month intervals for Aboriginal children in rural NSW	131 children participated in the program Majority (65.4%) received at least three fluoride varnish applications	1
Dimitropoulos et al., 2019 [8]	Explore experiences of school staff who implemented school toothbrushing programs in Aboriginal communities	Three focus groups conducted thematic analysis used to analyse transcripts	Four themes identified, including program need, routine, responses and sustainability	2
Kong et al., 2020 [9]	Experiences of Aboriginal health staff towards oral health care during pregnancy	Focus groups conducted with Aboriginal health staff	Four themes identified focusing on the role of Aboriginal health workers promoting maternal oral health. Results can be used to inform a model of oral healthcare for Aboriginal women during pregnancy	6
Dimitropoulos et al., 2020 [10]	Determine the impact of a community-led oral health promotion program for Aboriginal children in rural NSW that was implemented between 2016 and 2018	Impact evaluation conducted in 2018, including follow-up dental examination and interviewer-assisted questionnaires compared to baseline data collected in 2014	Reduction in dental caries, plaque scores and gingivitis. Mean number of teeth affected by dental caries was 4.13 (2018) compared to 5.31 (2014). Increases in positive oral hygiene behaviours	2
Campbell et al., 2015 [11]	Explore oral health care experiences of ACCHS in NSW	Online surveys and semi structured interviews were conducted within ACCHS in year	Oral health care provided by ACCHS is diverse and reflects the localised approaches they take to deliver primary health care. ACCHSs commonly face barriers in delivering oral health care and are under-acknowledged providers	6
Irving et al., 2017 [12]	Examine the views of children (and parents) who accessed a new dental service in rural NSW	Survey of the children who accessed this service was conducted between October and December 2014	High levels of oral pain were reported New dental service was easily accessible All respondents were happy with their treatment	6
Gwynne et al., 2017 [13]	Compare two models of oral health care for Aboriginal people including those living in rural NSW	Regression analysis was used to compare trends of dental weighted activity units of Model A (Fly-in-fly-out model) and Model B (community-led model)	The community-led model delivered more services for less financial resources	6
Dimitropoulos et al., 2019 [14]	A study protocol to enable dental assistants to apply fluoride varnish as part of a structured fluoride varnish program	A study protocol for a feasibility study for six Aboriginal dental assistants to undertake training in the application of fluoride varnish and apply fluoride varnish at 3-month intervals as part of a structured school fluoride varnish program	N/A	1
Skinner et al., 2020 [15]	Provide a costing for the scale-up of a child fluoride varnish program in New South Wales	The number of schools to be targeted as part of a national school fluoride varnish were described, and a costing method developed	Most of the costs of national school fluoride varnish program could be covered by the potential revenue from the Medicare Child Dental Benefits Schedule	1
Smith et al., 2018 [16]	To assess the effectiveness of a dental health education program ‘Smiles not Tears’	Aboriginal families with children across eight communities were invited to participate in the ‘Smile not Tears’ program. The program involved Aboriginal Health Workers delivering oral health messages to Aboriginal families with young children	97% (*n* = 104) of children in test group were caries-free compared to 65.9% (*n* = 54) of children in the control group	2
Dimitropoulos et al., 2018 [17]	Collaborate with Aboriginal communities in rural NSW to understand oral health needs and develop a targeted, community-owned oral health promotion program	Dental health status of Aboriginal children in 2014 was recorded Interviewer-assisted questionnaires conducted with children, parents/guardians and school staff	High level of dental caries and limited toothbrush and toothpaste ownership among children High consumption of sugar-sweetened beverages among children Limited oral health knowledge among parents/guardians School staff supportive of oral health promotion	2
Dimitropoulos et al., 2018 [18]	A study protocol to assess strategies to control dental caries in Aboriginal children	Strategies were to include Daily in-school toothbrushing Distribution of toothpaste and toothbrushes In-school and community dental health education Installation of refrigerated and filtered water fountains School water bottle program	N/A	2
Skinner et al., 2021 [19]	Identify key actions required to scale-up of school fluoride varnish programs	A workshop was held in Sydney (2018) with dental professionals from different jurisdictions and industry to discuss scale-up of fluoride varnish programs nationally	44 attendees attended the workshop There was strong support for the scale-up of fluoride varnish programs nationally Workshop recommendations included a standardised protocol, standardised legislation and support for dental assistants to apply fluoride varnish	1
George et al., 2018 [20]	Examined oral health behaviours and fluid consumption of young Aboriginal children in south-western Sydney	Parents of Aboriginal children aged 18–60 months completed an oral health survey	20% of parents/guardians were concerned about their child’s oral health 20% of children had seen a dentist 80% were brushing their teeth at least once daily High levels of bottle use were seen up to 30 months and consumption of sugary drinks	6
Irving et al., 2019 [21]	Evaluate improvements in oral health related quality of life (OHRQoL) of patients who received dentures from a novel mobile denture service	Aboriginal people who received a denture from the new service between July and December 2016 completed a survey at baseline and follow-up. A condensed version of the Oral Health Impact Profile Survey was used	28 people participated in the survey The effect of oral health on quality of life was improved in all measurement scores, particularly in psychological dimensions	6
Kong et al., 2021 [22]	Explore attitudes towards oral health among Aboriginal pregnant women and appropriate oral health promotion	Interviews were conducted with pregnant Aboriginal women and analysed thematically	Two themes were identified: Priority of oral health during pregnancy Importance of healthcare provider to support maternal oral health	5
Orr et al., 2021 [23]	Investigate use of the Child Dental Benefit Schedule (CDBS) among Aboriginal and non-Aboriginal children	CDBS Data for four financial years (2013–2014 and 2016–2017) was obtained. Logistic regression was used to estimate the odds of Aboriginal children using dental services through the CDBS	The use of the CDBS was lower among Aboriginal children. Aboriginal children were also less likely to access preventive dental services	6
Irving et al., 2017 [24]	Explore the experiences of dental clinicians who relocated to rural/remote communities to provide dental services in Aboriginal communities in Northern NSW	Semi-structured interviews were conducted with clinicians and reflective diaries kept. These were analysed qualitatively	Three themes were identified: Professional experience Personal growth Sense of achievement	5
Smith et al., 2015 [25]	To assess dental caries among young Aboriginal children	173 Aboriginal children in metropolitan, rural and remote NSW were examined	Dental caries among children were higher in remote locations when compared to rural and metropolitan areas. Children in remote areas had an average number of 3.5 teeth affected by dental caries compared to 1.5 for children living in rural areas	6
Gwynne et al., 2021 [26]	Compare the scope of practice for two models of dental service delivery in rural NSW	De-identified dental service records of two models of dental service delivery in rural NSW were clustered according to typical service groupings and analysed for the period 1 January 2014 to 31 December 2015	Model A (fly-in-fly-out) focused on more complex restorative dental care, whilst model B (community-led) provided a higher level of preventive care	6
Gwynne et al., 2016 [27]	Describe the steps taken to implement an oral health service for Aboriginal people in Central Northern NSW using the Collective Impact model	Partnerships were formed with the local community, schools and local health service providers to establish a dental service using portable dental equipment and graduate clinicians	The service provided reliable, high-quality care and built local community capacity	3
Skinner et al., 2021 [28]	Evaluation of a graduate oral health therapist program to support dental service delivery and oral health promotion in Aboriginal communities.	15 surveys completed by Graduate Oral Health Therapists participating in the Dalang Project between 2016–2018.	Participants reported learning to engage with Aboriginal communities and build cultural competence skills to provide culturally competent oral health services and develop oral health promotion programs.	2, 3

**Table 2 healthcare-10-00650-t002:** Scholarships for Aboriginal people undertaking tertiary studies in oral health.

Scholarship	Organisation	Location	Duration	Eligibility Criteria
The House Call Doctor Futures in Health Indigenous Scholarship [31]	House Call Doctor	National	1 Year	- Must identify as Aboriginal and/or Torres Strait Islander - 18 years of age or older - Australian citizens or permanent residents - An undergraduate or graduate student in the field of medicine or health in an Australian university
Grants for Indigenous Dental Students [32]	Australian Dental Association	National	1 Year	- Must identify as Aboriginal and/or Torres Strait Islander - Enrolled as full-time students in an Australian undergraduate or graduate entry dental degree immediately leading to registration as a dentist and must have completed at least the first year of that degree
Sydney Dental Hospital Indigenous Scholarship [33]	University of Sydney	NSW	1 Year	- Must identify as Aboriginal and/or Torres Strait Islander - Currently studying for a Bachelor of Oral Health or Doctor of Dental Medicine program at the University of Sydney
Albury Wodonga Aboriginal Health Service Scholarship [34]	Charles Sturt University	NSW	1 Year	- Must identify as Aboriginal and/or Torres Strait Islander - Have a good academic record - A resident of the Albury Wodonga Aboriginal Health service area

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
