# Peer review of "The Progress of the New South Wales Aboriginal Oral Health Plan 2014–2020: A Scoping Review"

_healthcare, 2022, doi:10.3390/healthcare10040650_

Round 1
Reviewer 1 Report
Dear Authors,
A scoping review cannot be assessed as a systematic review, meta-analysis or some experimental or observational study, therefore we can rely on your accuracy and reliability only.
Below these are some suggestions for You:
Affiliations:
- line 6; an incorrect abbreviation, rather than C.M.C should be C.M.S
Material and methods.
- line 78-80; who decided to include an article, in a case of mutual agreement - all of the authors?
- line 80-91; I am not sure the sentence is necessary there (it could be put in ‘Institutional Review Board Statement’, after conclusions, if it is relevant to your study)
Results:
- line 222-227; why italic type?
- line 255; the table should be placed near to the first time they are cited
Discussion:
- line 390-391; a (...) fountains?
- line 440; a full stop is missing
Conclusions:
- There is no necessity to keep conclusions, when discussion is not so long, but you can also preserve them
Best regards and good luck
Author Response
Response
Line 5 – this had been amended to CMS
Line79 – ‘All authors discussed the inclusion of articles’ has been added (now line 87)
Line 90 – re ethical consideration. We have decided to leave this sentence here to demonstrate that ethics approval was not required for this study.
Line 222-227 re italics – this has been changed to standard font
Line 255 - Table 1 has been moved to line 283 underneath where the table is first cited. Please note is now table 2
Line 390 – the letter ‘a’ has been deleted
Line 440 – Full stop has been added
Conclusion – thank you for this, however we have decided to keep the conclusion, keeping in the PRISMA checklist
Reviewer 2 Report
- Originality/Novelty: The question is original and well defined. The results provide an advance in current knowledge.
- Significance: The studies included in the scoping review are presented appropriately, and are significant for the theme studied. All conclusions are justified and supported by the results.
- Quality of Presentation: The article is written in an appropriate way. The data and analyses are presented appropriately
- Scientific Soundness: The study is correctly designed and technically sound. The methods is described with sufficient details to allow another researcher to reproduce the results.
- Interest to the Readers: The conclusions are interesting for the readership of the Journal.
- Overall Merit: The overall benefit to publishing this work resides in using the information as a model for another oral health interventions that reduce disparities between different groups of people.
- English Level: English language is appropriate and understandable.
Author Response
Thank you for your comments.
Reviewer 3 Report
The review shows the progress of New Wales Aboriginal oral health plan.
This is an interesting study. However, there are some issues. The paper needs to be revised.
Comments
First, the authors should follow the PRISMA-ScR Checklist. Second, the rationale is unclear.
ABSTRACT
- Pleas revise the abstract after the revision.
INTRODUCTION
- Please describe the rationale for the review in the context of what is already known.
- Please provide an explicit statement of the questions and objectives being addressed with reference to their key elements (e.g., population or participants, concepts, and context) or other relevant key elements used to conceptualize the review questions and/or objectives.
MATERIALS AND METHODS
- Please indicate whether a review protocol exists; state if and where it can be accessed (e.g., a Web address); and if available, provide registration information, including the registration number.
- Please add the name a university librarian.
- Please state the process for selecting sources of evidence (i.e., screening and eligibility) included in the scoping review.
- Please describe the methods of charting data from the included sources of evidence (e.g., calibrated forms or forms that have been tested by the team before their use, and whether data charting was done independently or in duplicate) and any processes for obtaining and confirming data from investigators. It is unclear what key words are and how “JS” judge the papers. Please add more detail methods.
- Please list and define all variables for which data were sought and any assumptions and simplifications made.
RESULTS
- Please add the flowchart.
- Please add a table in each subheading (3.1, 3.2, …and 3.6.) to summarize the results.
- If done, please present data on critical appraisal of included sources of evidence.
Author Response
Response
A PRISMA-SCR checklist has now been provided as a supplementary file.
ABSTRACT – the abstract has been revised to more accurately reflect the methodology
INTRODUCTION
- Lines 52-59, Rationale for the review in the context of what is already known has been strengthened.
- Lines 56-59, Explicit statement of the questions and objectives being addressed with reference to their key elements.
MATERIALS AND METHODS
- Line 66, A review protocol wasn’t registered for this study.
- Line 497, Name of the university librarian has been added.
- Line 70 Process for selecting sources of evidence (i.e., screening and eligibility) included in the scoping review has been revised.
- Lines 91-94The methods of charting data from the included sources of evidence has been revised.
- Lines 65-93 More detail has been provided in the methods.
RESULTS
- Flowchart added as supplementary file
- Table added to summarize the results (Table 1)
- Critical appraisal of included sources of evidence was not undertaken.
Round 2
Reviewer 3 Report
The review was overall improved. However, there are some issues. The paper needs to be revised.
Comments
The line number of answers does not match.
MATERIALS AND METHODS
- Please add the name a university librarian or initial in the text in addition to the acknowledgement.
- Please describe the methods of charting data from the included sources of evidence (independently or in duplicate) and any processes for obtaining and confirming data from investigators. It is still unclear. Furthermore, what is “six strategic directions”? Please add more detail methods.
RESULTS
- Please add the flowchart. There is no flowchart in the supplemental file.
Author Response
The initials of the librarian have been added to the text line: 61
The six strategic directions have been listed in lines 66-78
The methods for charting data have been clarified in lines 88-92
We apolgise that the flowchart was not uploaded, it has now been uploaded as a supplementary file.